# Transcriptomics-informed large-scale cortical model captures topography of pharmacological neuroimaging effects of LSD

Joshua B Burt[1], Katrin H Preller[2,3], Murat Demirtas[3], Jie Lisa Ji[1,4], John H Krystal[3], Franz X Vollenweider[5], Alan Anticevic[3,4], John D Murray[1,3,4]*

[1]Department of Psychiatry, Yale University, New Haven, United States; [2]Pharmaco-Neuroimaging and Cognitive-Emotional Processing, Department of Psychiatry, Psychotherapy and Psychosomatics, University Hospital for Psychiatry Zurich, Zurich, Switzerland; [3]Department of Psychiatry, Yale University School of Medicine, New Haven, United States; [4]Interdepartmental Neuroscience Program, Yale University, New Haven, United States; [5]Neuropsychopharmacology and Brain Imaging, Department of Psychiatry, Psychotherapy and Psychosomatics, University Hospital for Psychiatry Zurich, Zurich, Switzerland

**Abstract** Psychoactive drugs can transiently perturb brain physiology while preserving brain structure. The role of physiological state in shaping neural function can therefore be investigated through neuroimaging of pharmacologically induced effects. Previously, using pharmacological neuroimaging, we found that neural and experiential effects of lysergic acid diethylamide (LSD) are attributable to agonism of the serotonin-2A receptor (Preller et al., 2018). Here, we integrate brain-wide transcriptomics with biophysically based circuit modeling to simulate acute neuromodulatory effects of LSD on human cortical large-scale spatiotemporal dynamics. Our model captures the inter-areal topography of LSD-induced changes in cortical blood oxygen level-dependent (BOLD) functional connectivity. These findings suggest that serotonin-2A-mediated modulation of pyramidal-neuronal gain is a circuit mechanism through which LSD alters cortical functional topography. Individual-subject model fitting captures patterns of individual neural differences in pharmacological response related to altered states of consciousness. This work establishes a framework for linking molecular-level manipulations to systems-level functional alterations, with implications for precision medicine.

*For correspondence: john.murray@yale.edu

## Introduction

What are the respective roles of structure and physiology in shaping the spatiotemporal dynamics of networked neural systems? Areal differences in gene transcription, cellular architecture, and long-range connectivity patterns have been linked to areal differences in specialization of cortical function (*Felleman and Van Essen, 1991*; *Hilgetag et al., 2016*; *Burt et al., 2018*; *Huntenburg et al., 2018*; *Demirtaş et al., 2019*). Yet it remains unclear how embedded neurophysiological changes impact functional network organization and structure-function relationships. Non-invasive neuroimaging of pharmacologically induced changes in brain function permits selective, transient, and in vivo perturbation of physiology, while preserving brain structure such as inter-areal axonal projections. Pharmacological neuroimaging therefore provides a tractable and valuable testbed for investigating the roles of neurophysiological properties in shaping patterns of large-scale brain function.

The central role of physiological state in shaping functional brain dynamics and influencing human consciousness and behavior is supported by neuroimaging studies of the acute functional disturbances induced by psychopharmacological compounds. Serotonergic hallucinogens, in particular, produce rapid and profound alterations of consciousness that are linked to widespread, stereotyped patterns of functional network disruption (*Tagliazucchi et al., 2014*; *Tagliazucchi et al., 2016*; *Lord et al., 2018*; *Muthukumaraswamy et al., 2013*; *Atasoy et al., 2017*; *Preller et al., 2018*; *Preller et al., 2020*). Recently, our group has shown that lysergic acid diethylamide (LSD)-induced disruptions of blood oxygen level-dependent (BOLD) functional connectivity (FC) and concomitant changes in consciousness are extinguished by pre-treatment with ketanserin, a selective serotonin-2A (5-HT$_{2A}$) receptor antagonist (*Preller et al., 2018*). This result suggests that the 5-HT$_{2A}$ receptor plays a critical role in LSD's mechanism of action (*Vollenweider and Preller, 2020*; *Holze et al., 2021*). *Preller et al., 2018* analyzed regional changes in mean BOLD FC, called global brain connectivity (GBC), which is a graph-theoretic statistic that can be interpreted as a measure of functional integration (*Cole et al., 2010*; *Anticevic et al., 2014*). We found that the regional topography of LSD-induced changes in GBC is aligned with the topography of regional expression levels of HTR2A, the gene which encodes the 5-HT$_{2A}$ receptor (*Preller et al., 2018*). However, the circuit mechanism through which LSD alters cortical GBC topography remains unclear.

One approach to bridge this mechanistic gap is to develop biophysically based models of large-scale brain activity that incorporate key features of neuronal and synaptic dynamics (*Breakspear, 2017*; *Deco et al., 2011*). These models, which are grounded in human neurobiology, can be first calibrated to healthy-state data and then systematically perturbed through biophysically interpretable parameters. In doing so, selective manipulations at the synapse can be linked to their manifestations at empirically resolvable scales (*Shine et al., 2021*) – for instance, at the length scale and timescales probed by functional magnetic resonance imaging (fMRI). Moreover, recent advances in high-throughput transcriptomics have led to rich genome-wide atlases of gene expression levels mapped across the human brain (*Hawrylycz et al., 2012*; *Hawrylycz et al., 2015*). Insofar as protein levels increase with the number of gene transcripts in a region (*Liu et al., 2016*), gene expression maps provide a principled way to infer the spatial distribution of proteins of interest, for example, drug targets. Gene expression maps therefore provide a means to simulate the regionally heterogeneous impacts of a drug on local circuit properties (*Murray et al., 2018*).

We extended this computational modeling approach to generate mechanistic insight into the topography of LSD-induced GBC alterations (*Preller et al., 2018*). GBC is pharmacologically elevated in sensory cortex, especially in visual cortex, and reduced in association cortex (*Preller et al., 2018*). Moreover, GBC changes correlate significantly with changes in subjects' experience of consciousness, as determined by validated psychometric instruments (*Studerus et al., 2010*; *Preller et al., 2018*). Here, we investigated potential circuit mechanisms underlying these effects by extending a previously validated model of large-scale neural dynamics (*Deco et al., 2013*; *Deco et al., 2014*; *Demirtaş et al., 2019*). We found that our model accurately captures the spatial topography of LSD-induced GBC changes. Our findings suggest that the distribution of 5-HT$_{2A}$ receptors is critical for generating the cortical topography of functional disruptions, and that neural gain is preferentially modulated on cortical pyramidal neurons, consistent with known neurobiology. Fitting the model at the individual-subject level, we found that the model captures patterns of individual differences in drug response that predict altered states of consciousness. Our work therefore advances pharmacological modeling from global effects at the group level to spatially heterogeneous effects in individuals. Broadly, this work establishes a flexible conceptual framework for linking mechanistic molecular hypotheses to human neuroimaging markers.

## Results

To investigate the circuit mechanism through which LSD alters cortical GBC topography, we extended a previously validated biophysically based model of large-scale neural circuit dynamics (*Wong and Wang, 2006*; *Deco et al., 2013*; *Demirtaş et al., 2019*; *Figure 1*). The model comprises many local microcircuits, each consisting of coupled excitatory and inhibitory neuronal populations. Population dynamics are driven by recurrent synaptic interactions that are governed by

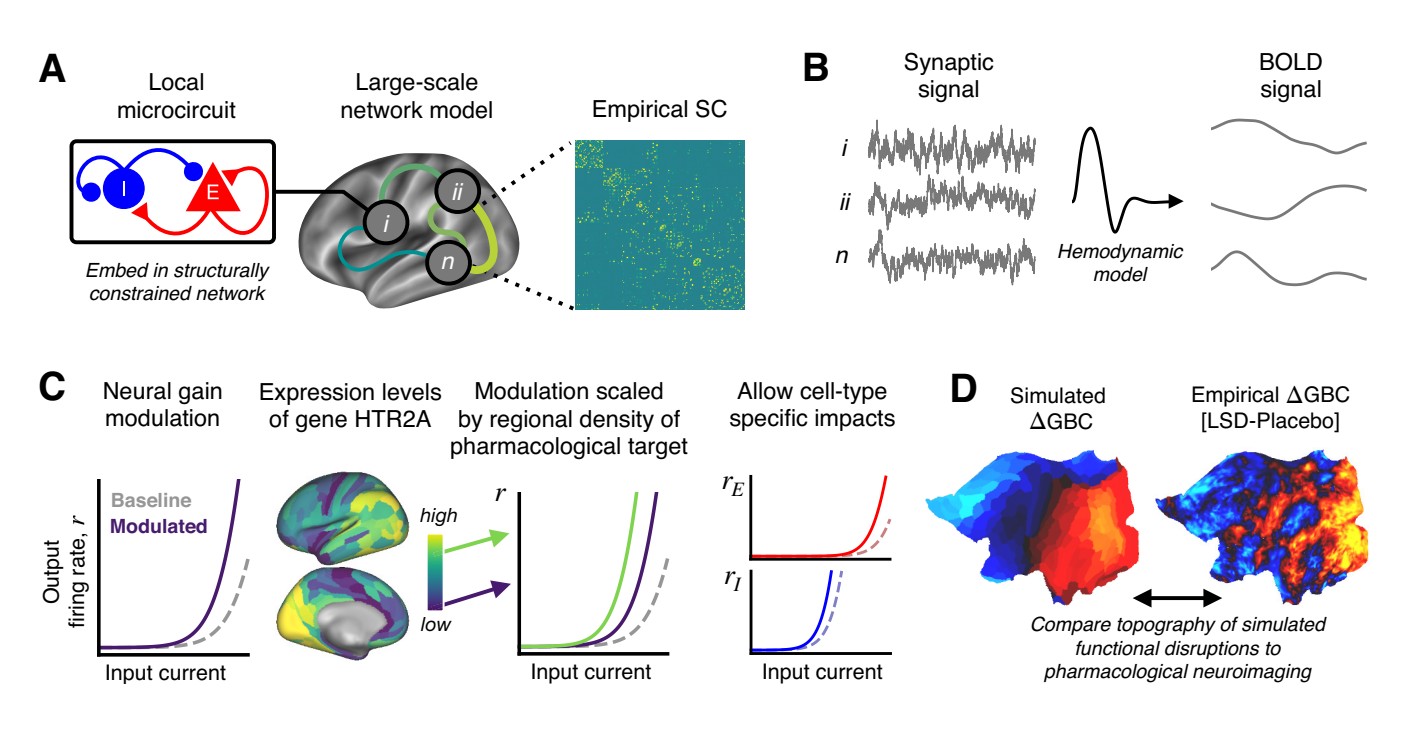

**Figure 1.** Schematic overview of the biophysical modeling framework. (**A**) Each node in the large-scale model represents a cortical microcircuit comprised of recurrently coupled excitatory (*E*) and inhibitory (*I*) neuronal populations. The model includes one node for each of the 180 left-hemispheric parcels in the Human Connectome Project's Multi-Modal Parcellation (MMP1.0). Nodes interact through structured long-range excitatory projections, the strengths of which are constrained by a diffusion magnetic resonance imaging (MRI)-derived structural connectivity (SC) matrix. (**B**) Simulated synaptic activity in each node is transformed to a simulated blood oxygen level-dependent (BOLD) signal using the Balloon-Windkessel model of the hemodynamic response. (**C**) Lysergic acid diethylamide (LSD)'s effect on cortical microcircuitry is modeled as a modulation of neural gain due to serotonin-2A (5-HT$_{2A}$) receptor agonism by the LSD molecule. The degree to which neural gain is modulated within an area is scaled in proportion to the regional expression level of HTR2A, the gene which encodes the 5-HT$_{2A}$ receptor protein. Gain curves of the excitatory and inhibitory neuronal populations are modulated independently, permitting cell-type specific effects. (**D**) Global brain connectivity (GBC), a graph-theoretic functional measure, is dramatically altered following LSD administration. The functional MRI (fMRI)-derived map of the change in GBC ($\Delta$GBC) under LSD, relative to placebo, specifies the target model output. To simulate brain function in the LSD and placebo drug conditions, we simulate GBC maps with and without gain modulation, respectively. We compute the difference between the model GBC maps to construct a simulated $\Delta$GBC map. Quantitative comparisons between empirical and model $\Delta$GBC maps determine how well the model captures the topography of LSD-induced functional disruptions.

The online version of this article includes the following figure supplement(s) for figure 1:

**Figure supplement 1.** In the human brain, HTR2A is predominately expressed in cortical pyramidal neurons.

neurophysiologically interpretable parameters. Our model consists of 180 interconnected nodes, each of which represents one left-hemispheric cortical parcel in the Human Connectome Project's (HCP's) Multi-Modal Parcellation (MMP1.0) (*Glasser et al., 2016*). Note that we model only one cortical hemisphere, as the pharmacological fMRI data informing the model are approximately bilaterally symmetric (*Preller et al., 2018*), and the gene expression data are sampled from the left hemisphere but show bilateral correspondence (*Hawrylycz et al., 2015*). Nodes in the network interact through structured long-range excitatory projections. The relative strengths of these projections are informed by a diffusion MRI-derived, group-averaged (N = 339) structural connectivity (SC) matrix (*Figure 1A*; *Van Essen et al., 2013*; *Demirtaş et al., 2019*). The SC matrix is row-wise normalized such that total long-range inputs to each node are balanced and independent of parcel size. To evaluate model outputs against fMRI-derived data, simulated synaptic activity in each node is transformed into an observable BOLD signal using the Balloon-Windkessel model for the hemodynamic response (*Friston et al., 2003*; *Stephan et al., 2007*; *Deco et al., 2014*; *Demirtaş et al.,*

*2019*; *Figure 1B*). We henceforth refer to this hemodynamically coupled neural circuit model as the 'baseline' or 'unperturbed' model.

To simulate the action of LSD in human cortex, we systematically perturb the baseline model. In earlier empirical analyses, we found that LSD's influence on brain function and on behavior is primarily mediated through its agonist activity at the 5-HT$_{2A}$ receptor (*Preller et al., 2018*). Stimulation of 5-HT receptors alters the response properties and membrane excitability of mammalian cortical neurons, such that firing rates are selectively enhanced for strongly depolarizing currents – that is, such that neural gain is enhanced (*Araneda and Andrade, 1991*; *Andrade, 2011*; *Zhang and Arsenault, 2005*). This 5-HT receptor-mediated enhancement or modulation of gain is thought to be mediated specifically by the 5-HT$_{2A}$ receptor (*Zhang and Arsenault, 2005*). We therefore simulate the action of LSD by manipulating neural gain equations in the model (*Figure 1C*). Neural gain in the model is defined by a non-linear expression of the form $r = f(aI)$ (*Equation 2*), which relates pre-synaptic current $I$ to post-synaptic firing rate $r$ by the non-linear function $f$ with scalar gain parameter $a$. We mathematically express gain modulation as a fractional change in $a$, that is, $a \to (1 + \delta)a_0$, where $a_0$ is the unperturbed parameter value and $\delta$ determines the magnitude of modulation.

To account for the heterogeneous spatial distribution of 5-HT$_{2A}$ receptor density across cortical areas, gain modulation in each node is scaled in proportion to the local expression level of HTR2A – the gene which encodes the 5-HT$_{2A}$ receptor – using transcriptomic data from the Allen Human Brain Atlas (AHBA) (*Figure 1—figure supplement 1*; *Hawrylycz et al., 2012*; *Burt et al., 2018*). In cortex, 5-HT$_{2A}$ receptors are present on glutamatergic projection neurons and GABAergic interneurons (*Mengod et al., 2010*; *Burnet et al., 1995*; *Santana et al., 2004*). We introduce an additional degree of freedom into the model such that gain modulation can differentially impact the excitatory and inhibitory neuronal populations, thereby not making a priori assumptions as to the cell-type specificity of 5-HT$_{2A}$-mediated gain modulation (*Figure 1C*). Specifically, the gain parameter $a$ of neuronal population $p$ in brain region $i$ is given by $a_i^p = (1 + h_i\delta^p)a_0^p$, where $h_i$ is proportional to the expression level of gene HTR2A in the $i$ th region. For brevity, in what follows we drop the subscript $i$ and superscript $p$ notation. Note that the scalar gain parameter $a$ depends linearly on $\delta$, and the gain function $r = f(aI)$ scales non-linearly with $a$; thus, the gain functions scale non-linearly with $\delta$.

## Quantifying functional organization with GBC maps

Functional organization of cortical dynamics, in the model and in the data, can be operationalized by constructing cortical GBC maps, for comparison to our prior empirical findings (*Preller et al., 2018*). The change in GBC induced by LSD provides a topographic map across cortical regions of the mean effect on FC, which we have previously compared to the topography of the HTR2A gene expression map (*Preller et al., 2018*). The first step in computing a GBC map is to construct an FC matrix, defined as the matrix of Pearson correlation coefficients between pairs of brain regions' BOLD signal time traces. We apply the Fisher r-to-Z transformation to off-diagonal elements of this matrix such that values are approximately normally distributed. Off-diagonal elements are then averaged along either rows or columns (as the FC matrix is symmetric), yielding a map of GBC which consists of a single scalar value for each region.

As reported in *Preller et al., 2018*, GBC maps were constructed from resting-state BOLD fMRI scans for 24 study participants. Subjects served as their own controls. GBC maps were constructed for each subject in both drug conditions (placebo and LSD). The group-averaged contrast map of the change in GBC for LSD vs. placebo drug conditions – henceforth referred to as the ΔGBC map – specified the target model output (*Figure 1D*). In other words, we sought to recapitulate in our model the pattern of functional alterations indicated by changes in cortical GBC topography.

Empirically, the group-averaged ΔGBC map reveals widespread LSD-induced hyper-connectivity of sensory-somatomotor regions, and hypo-connectivity of association regions (*Preller et al., 2018*). In the models, we generate a GBC map for both the unperturbed and neuromodulated models, corresponding to models for the placebo and LSD conditions, respectively (described in more detail below). From these maps, we compute a model ΔGBC map which is then quantitatively compared to the empirical ΔGBC map (*Figure 1D*).

## Model captures the spatial topography of LSD-induced changes in cortical GBC

The operating point of the unperturbed model was first calibrated such that it best captured empirical FC in the placebo condition. This was achieved by performing a one-dimensional grid search over the value of a global long-range coupling parameter, which uniformly scales the strengths of all long-range connections between nodes (*Deco et al., 2014*). We determined the value of this coupling parameter that yielded the greatest Spearman rank correlation between the off-diagonal elements of the simulated and empirical FC matrices. The coupling parameter was subsequently fixed at this value. The model FC matrix was then used to compute a baseline model GBC map – our computational analog of the empirical GBC map in the placebo condition.

Next we performed a two-dimensional grid search over the two free model parameters which set the strength of gain modulation on excitatory and inhibitory neurons ($\delta^E$ and $\delta^I$), such that a modulation of gain by 1% corresponds to $\delta = 0.01$ (*Figure 2A*). For each combination of parameters (i.e., at each point on the parameter space grid), we updated the simulated gain functions and re-computed the system's stable fixed point. This now-perturbed system is used to generate a new FC matrix, from which we again compute a model GBC map – this time, our computational analog of the empirical GBC map in the LSD condition. For each combination of free parameters, we then computed a model ΔGBC map by calculating the difference between the perturbed and unperturbed model GBC maps. To evaluate the quality of the model for each set of parameters, we evaluated the statistical similarity of each model ΔGBC map to the empirical ΔGBC map. We quantified similarity by computing the 'loading' of the model ΔGBC map onto the empirical ΔGBC map, which we define as the Cartesian dot product between the two ΔGBC maps (expressed as vectors), divided by the squared norm of the empirical ΔGBC map (the natural scale in the quantity). This metric was chosen specifically for its sensitivity to both the topography and magnitude of the model perturbation.

We found that the model ΔGBC map loaded most strongly onto the empirical ΔGBC map across a quasi-degenerate range of parameter values (*Figure 2A*, off-diagonal yellow band). The location of this regime suggests that neural gain is preferentially modulated on excitatory pyramidal neurons, which nonetheless remain in a specific ratio with inhibitory interneurons. To investigate this further, we next characterized the relationship between changes in model excitatory-to-inhibitory (*E/I*) balance and model-empirical loading. For each combination of gain modulation parameters (i.e., at each location on the two-dimensional grid in parameter space), we computed the model's *E/I* firing rate ratio, relative to its unperturbed value, where we define *E/I* ratio as the ratio of the mean excitatory firing rate (computed across nodes) to the mean inhibitory firing rate. We find that model-empirical loadings, when plotted as a function of fractional change in *E/I* ratio, collapse to form an approximately one-dimensional curve which peaks at positive shifts in *E/I* ratio (*Figure 2B*). This one-dimensional collapse suggests that gain modulation-induced changes in *E/I* balance are the primary mechanism through which GBC topography is altered under LSD. Moreover, the fact that the peak occurs for positive shifts in *E/I* balance suggests that LSD has a net-excitatory effect on cortical activity.

The model generates a strong functional perturbation that is aligned with the data, but does it also capture fine-grained functional network-specific and parcel-level effects? Empirically, LSD exhibits bidirectional effects across sensory and association brain networks, inducing hyper-connectivity of constituent brain regions in sensory and somatomotor networks, while inducing hypo-connectivity of regions in associative networks (*Preller et al., 2018*). To investigate network effects in the model, we set the gain modulation parameters to the values that yielded the greatest model-empirical loading (indicated by the black star in *Figure 2A*). We then characterized the distribution of model ΔGBC values across parcels within each functional network defined in the Cole-Anticevic Brain Network Parcellation (CAB-NP) (*Figure 2C*; *Ji et al., 2019*). We found that empirical network specificity of GBC changes was recapitulated in the model (*Figure 2D–E*). This finding indicates that the strong model-empirical loading is not driven by select functional networks or by a small subset of all cortical parcels. In fact, at the level of parcels we find a remarkably strong spatial correspondence between ΔGBC topographies in the model and data ($r_s = 0.73$; p $< 10^{-5}$; Spearman rank correlation) (*Figure 2F*). Note that brain map values, here and in subsequent figures, are plotted on a flattened (i.e., unfolded) representation of the cortical surface, to better illustrate maps' contiguous spatial topographies. This statistical association was significantly stronger than the association between the

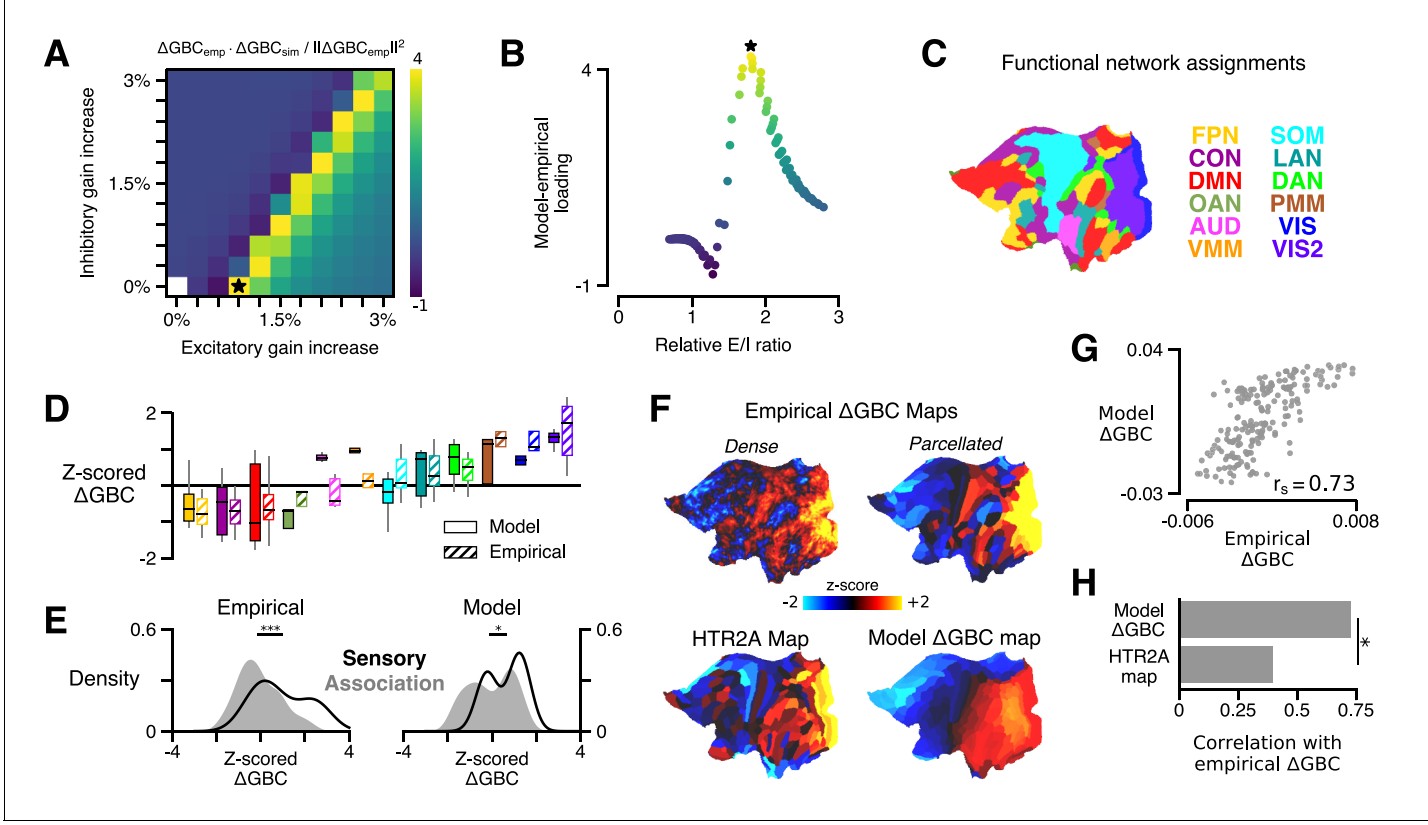

**Figure 2.** HTR2A-mediated excitatory gain modulation captures effects of lysergic acid diethylamide (LSD) on human cortical global brain connectivity (GBC) topography. (**A**) Two-dimensional grid search over the two free model parameters. Parameters govern the gain modulation of inhibitory and excitatory neuronal populations in the model. Model-empirical loading – the quantity shown in the heatmap – is defined as the dot product between the empirical change in GBC (ΔGBC) map and a model ΔGBC map, normalized by the squared norm of the empirical ΔGBC map. Loading is maximized for the combination of parameters indicated by the black star. (**B**) At each point on the grid (i.e., for each combination of gain-modulatory parameters), we computed the excitatory-to-inhibitory (E/I) firing rate ratio, expressed in terms of its unperturbed value. This amounts to computing $r'/r$, where $r'$ is the E/I ratio in the model with gain modulation, and $r$ denotes the E/I ratio in the model without gain modulation. E/I ratio is defined in the model as the ratio of the mean excitatory firing rate (computed across nodes) to the mean inhibitory firing rate. (**C**) Functional network assignments for each cortical parcel are determined by the Cole-Anticevic Brain Network Parcellation (CAB-NP): ventral multi-modal (VMM), language (LAN), dorsal attention (DAN), posterior multi-modal (PMM), primary visual (VIS), secondary visual (VIS2), frontoparietal (FPN), cingulo-opercular (CON), default mode (DMN), orbito-affective (OAN), auditory (AUD), and somatomotor (SOM) networks. Network colors mirror (*Ji et al., 2019*). (**D**) Functional network-level comparisons between simulated (solid) and empirical (striped) z-scored ΔGBC map values. Box plots mark the median and inner quartile ranges for parcels in each network, and whiskers indicate the 95% confidence interval. (**E**) Distributions of z-scored ΔGBC map values across cortical parcels in primary sensory networks (AUD, VIS, VIS2, SOM; black line with no fill) and association networks (gray fill with no line). Endpoints of the horizontal black lines (top) correspond to the distributions' means. Distributions significantly differ empirically ($p < 10^{-4}$; spatial autocorrelation-preserving surrogate map test) and in the model ($p = 0.02$; spatial autocorrelation-preserving surrogate map test). (**F**) Spatial topographies of the dense and parcellated empirical ΔGBC maps; the strongest-loading model ΔGBC map; and the HTR2A gene expression map. Maps are portrayed on flattened (unfolded) representations of the cortical surface. (**G**) Scatter plot illustrating the parcel-wise relationship between the strongest-loading model ΔGBC map and the empirical ΔGBC map. (**H**) Spearman rank correlations between the empirical ΔGBC map and: (i) the strongest-loading model ΔGBC map; and (ii) the HTR2A expression map. Empirical ΔGBC topography is better explained by the dynamical model than by the HTR2A map ($p < 0.05$; test for dependent correlations).

The online version of this article includes the following figure supplement(s) for figure 2:

**Figure supplement 1.** Effect of global signal regression (GSR) on model performance.

HTR2A gene expression map and the empirical ΔGBC map (p = 0.011; test for difference between dependent correlations) (*Figure 2H*). Heterogeneous physiological responses to LSD thereby play a critical role in shaping the cortical topography of LSD-induced GBC changes.

Global signal regression (GSR) remains a debated pre-processing step that is frequently included in fMRI analyses (*Murphy and Fox, 2017*). *Preller et al., 2018* extensively characterized these data, with and without GSR applied. There, a statistically significant brain-behavior relationship between

the neural and experiential effects of LSD was observed when – and only when – GSR was performed. Moreover, GSR was found to substantially alter the topography of the empirical ΔGBC map: when GSR was not used, the spatial correspondence of the ΔGBC map with the HTR2A expression map is dramatically attenuated (*Preller et al., 2018*). Here, we find that when GSR is not performed, maximal model-empirical loading and Spearman rank correlation between the model and empirical ΔGBC maps are both reduced significantly (*Figure 2—figure supplement 1*). This finding further supports the notion that GSR attenuates non-neural components or artifacts which otherwise obscure meaningful neuronal effects in pharmacological fMRI signals.

## Receptor topography plays a critical role in shaping large-scale functional effects

In addition to stimulating 5-HT$_{2A}$ receptors, LSD exerts predominately agonistic activity at (5-HT)-2C, -1A/B, -6, and -7 receptors, as well as dopamine D1 and D2 receptors (*Nichols, 2004*; *Passie et al., 2008*). Molecular signaling across these disparate receptor subtypes would impact neuronal physiology differently; however, these neuromodulatory systems are known to be important regulators of cortical gain (*Thurley et al., 2008*; *Ferguson and Cardin, 2020*). We therefore asked in our model whether modulation of gain via these other receptor subtypes also explains the spatial topography of cortical GBC changes. To test this, we repeated our grid search over model parameters, except that instead of modulating gain in proportion to HTR2A expression levels, we modulated gain in proportion to the expression level of, in turn: serotonergic genes HTR1A, HTR2C, and HTR7, and dopaminergic genes DRD1 and DRD2 (excluding 5-HT$_{1B}$ and 5-HT$_6$ encoding genes for lack of reliable transcriptomic data in the AHBA). The result of this process is illustrated in *Figure 3A*. Of the receptor-encoding genes that we tested, using the HTR2A map in the model yields the maximal model-empirical loading and Spearman rank correlation between empirical and simulated ΔGBC maps (*Figure 3B*). These findings highlight the importance of the spatial distribution of drug targets in mediating their large-scale functional effects.

We examined whether a strong model-empirical loading could be simply explained by general statistical properties of the HTR2A map, rather than its specific topographic pattern. To quantify the significance of the HTR2A map's topography, we sought to establish statistical expectations for model-empirical loading under an appropriate null hypothesis. A widely adopted approach to quantifying significance for a complex, model-generated statistic (such as model-empirical loading) is to perform a non-parametric permutation test: by randomly permuting the brain map and re-generating a quantity of interest, one can construct a null distribution of expected outcomes due to random chance (*Deco et al., 2018*). When permutation tests of this type are applied to brain maps, the implicit null hypothesis is that any brain map with the same distribution of values but with a random topography is statistically likely to have produced a comparable or more extreme effect.

One important problem with this approach is that spatial autocorrelation – a characteristic statistical property of brain maps – is necessarily destroyed by random spatial permutations. Spatial autocorrelation is fundamentally important for two reasons: (i) it violates the assumption that samples are independent or exchangeable, an assumption which underlies many common statistical tests (including the permutation test); and (ii) this violation dramatically inflates p-values in studies of brain maps (*Burt et al., 2020*). In light of this limitation, here we leverage a recent method to generate autocorrelation-preserving surrogate brain maps (*Burt et al., 2020*). By construction, these surrogate brain maps have randomized topographies but a fixed spatial autocorrelation (*Figure 3C*). Building a null distribution from these autocorrelation-preserving brain maps facilitates a test that controls for this important property.

To perform such a test, we repeatedly redefined gain functions in the model, each time by modulating gain in proportion to regional values of a random spatial autocorrelation-preserving surrogate brain map. Each surrogate map was constructed to have the same spatial autocorrelation structure as the HTR2A map. For each surrogate map, we regenerated a model ΔGBC map, from which we computed a model-empirical loading. The resulting null distribution of model-empirical loadings was then used to evaluate the significance of the HTR2A map's specific topography (*Figure 3D*). We found that modulating gain in proportion to HTR2A expression levels produces a statistically significant loading (p = 0.008). The null distribution further reveals that none of the alternative gene expression maps reach significance of p < 0.05. These findings further implicate the 5-HT$_{2A}$ receptor

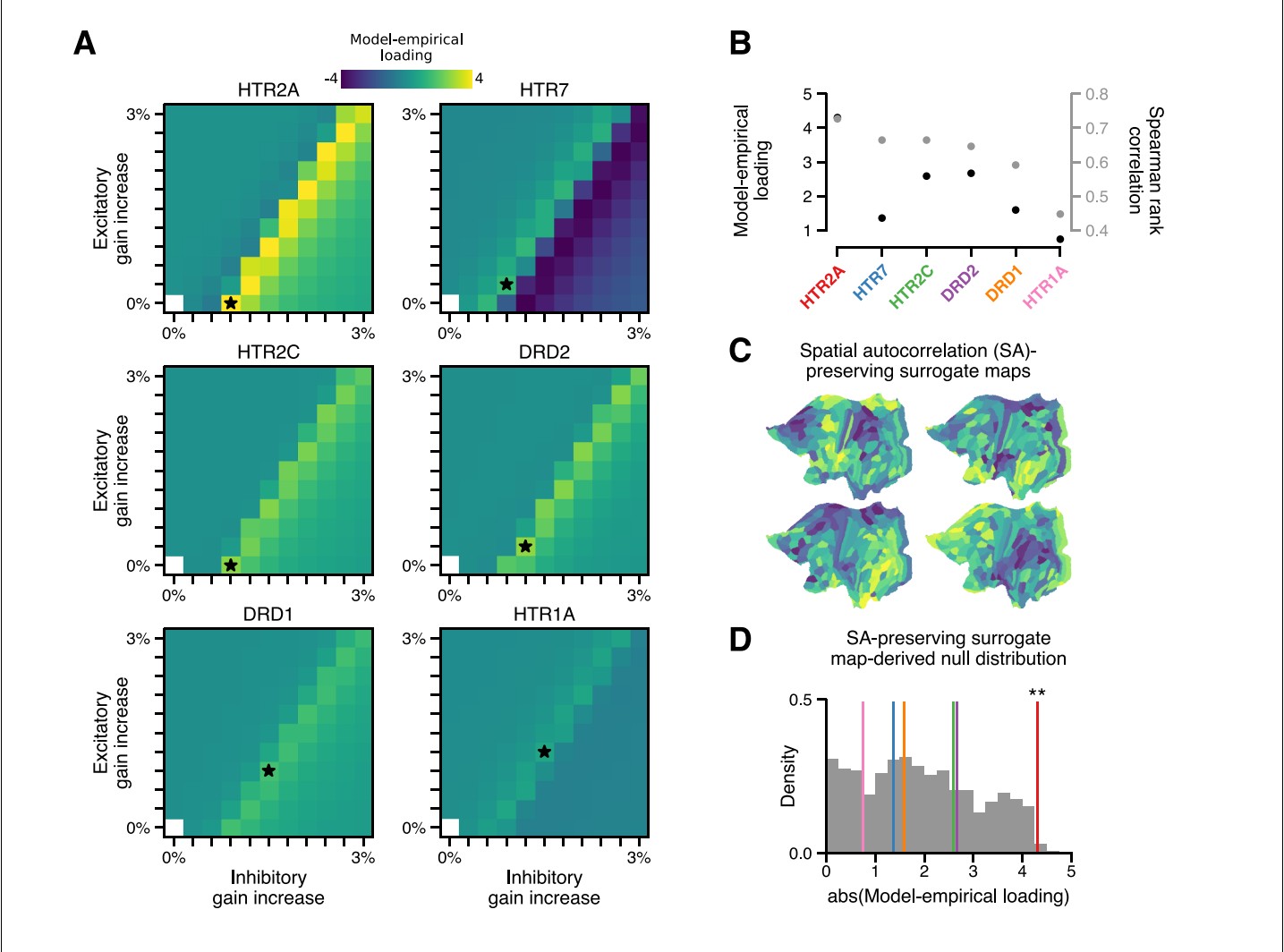

**Figure 3.** The topography of lysergic acid diethylamide (LSD)-induced cortical global brain connectivity (GBC) changes is specifically attributable to the spatial distribution of 5-HT$_{2A}$ receptors, as indexed by the HTR2A map. (A) Two-dimensional grid searches over free model parameters. For each heatmap, gain modulation is scaled in proportion to regional expression levels of different serotonergic (HTR) and dopaminergic (DRD) receptor-encoding genes, each of which is agonized by LSD. Black stars indicate maximal model-empirical loadings for each heatmap. (B) Model-empirical loading (left axis) and Spearman rank correlation (right axis) are greatest when gain is modulated by regional expression levels of HTR2A. Model change in GBC (ΔGBC) maps used in this analysis were generated using the gain-modulatory parameters that maximized model-empirical loadings, as indicated on each heatmap. (C,D) Following *Burt et al., 2020*, we generate surrogate brain maps with randomized spatial topographies that, by construction, exhibit spatial autocorrelation that has been matched to that of the HTR2A map. These spatial autocorrelation-preserving surrogate brain maps are used to construct a null distribution of the expected magnitude of model-empirical loading under random chance. Each sample in the null distribution (gray; *N* = 1000) was constructed by modulating gain in proportion to the values in a random spatial autocorrelation-preserving surrogate brain map, then re-computing model-empirical loading. Colored lines correspond to the different receptor-encoding genes (as reported in panel B). ** p < 0.01.

in LSD's mechanism of action, while simultaneously acting as a statistical control for the model complexity.

## Model captures experientially relevant modes of neural variation

Our model captures cortical GBC alterations at the group level, but is it expressive enough to capture individual differences? To characterize individual variation – first, in the empirical data – we performed a principal components analysis (PCA) on subjects' individual ΔGBC maps. We focused on the leading principal component (PC1) of ΔGBC variation (*Figure 4A*). By definition, PC1

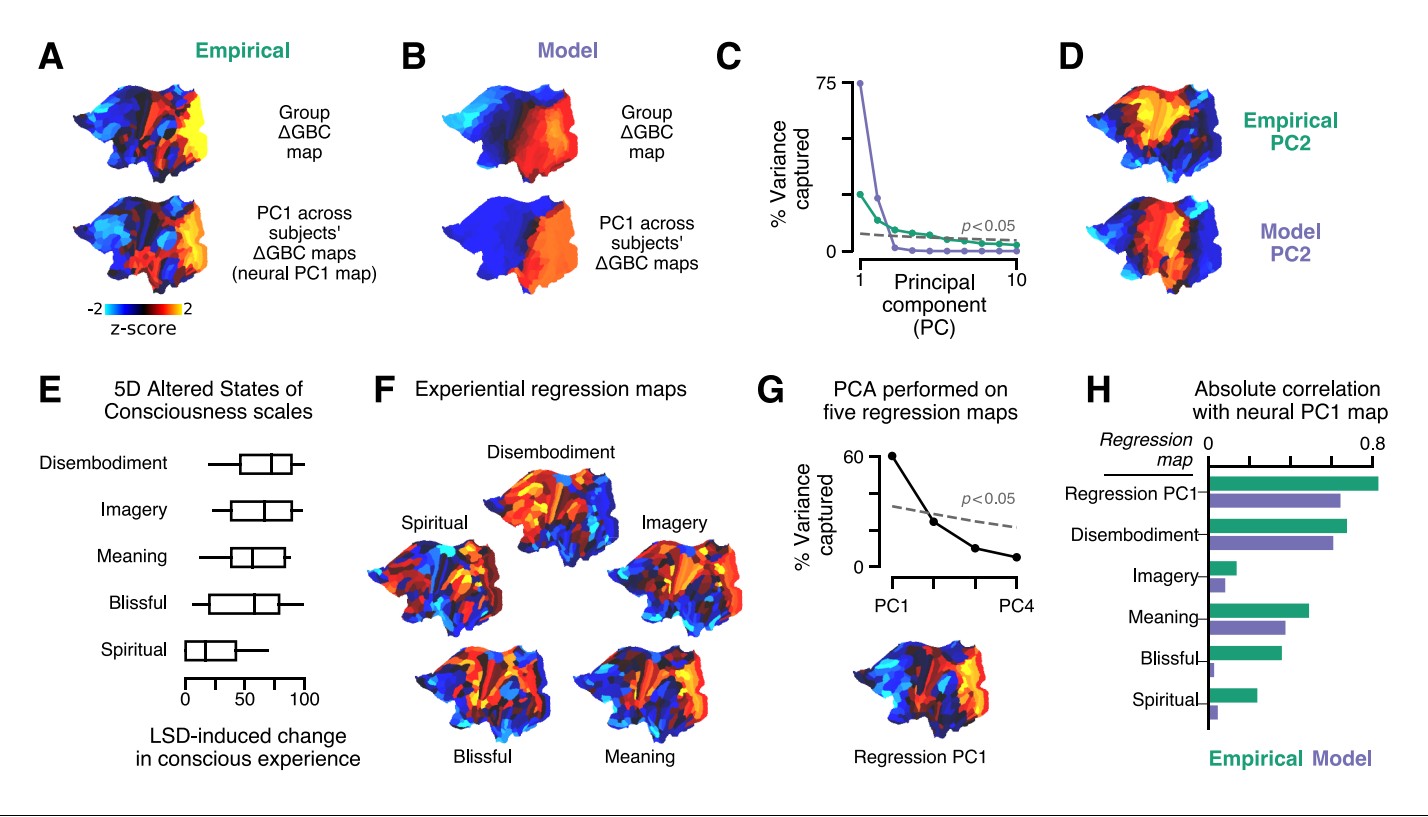

**Figure 4.** Model fits to individual subjects capture experientially relevant modes of neural variation. (A–D) Characteristic patterns of inter-individual variability in cortical change in global brain connectivity (ΔGBC) topography. (A) The empirical group-averaged ΔGBC map (top), and the leading principal component (PC1) computed across 24 subjects' empirical ΔGBC maps (bottom). (B) The model ΔGBC map fit to the group-averaged data (top), and the PC1 computed across 24 subjects' model ΔGBC maps (bottom). (C) Empirical (green) and model (purple) PC variance spectra. The first five empirical PCs and the first two model PCs survived permutation testing (p < 0.05, 1000 permutations). (D) Empirical and model PC2 maps. (E–H) Linking individual differences in ΔGBC to individual differences in lysergic acid diethylamide (LSD)-induced alterations of consciousness. (E) Changes in subjects' conscious experience under LSD relative to placebo as determined by the five-dimensional (5D) altered states of consciousness (5D-ASC) questionnaire: disembodiment, elementary imagery, changed meaning of percepts, blissful state, and spiritual experience. Box plots mark the median and inner quartile ranges for each scale, and whiskers indicate the 95% confidence interval. Positive values indicate an increase under lysergic acid diethylamide (LSD). (F) Experiential regression maps are constructed by performing linear regressions, across subjects, between changes in an experiential score (the target variable) and changes in GBC within a single parcel (the predictor variable). For each 5D-ASC scale, we performed 180 regressions – one per parcel – across 24 subjects. Brain maps illustrate the first-order regression coefficients. Experiential regression maps therefore illustrate patterns of GBC variation that predict experiential variation. (G) We performed principal components analysis (PCA) on the five experiential regression maps to derive experiential regression map principal components (PCs). *Top*: variance spectrum for the experiential regression map PCs. PC1 survives permutation testing (p < 0.05, 1000 permutations). *Bottom*: experiential regression map PC1, which captures 60% of variance across experiential regression maps. (H) Spearman rank correlations between experiential regression maps and neural PC1 maps.

The online version of this article includes the following figure supplement(s) for figure 4:

**Figure supplement 1.** Empirical change in global brain connectivity (ΔGBC) maps for each subject.

corresponds to the spatial map that (linearly) captures maximal variance in ΔGBC across subjects. We found that PC1 correlates strongly with the group-averaged ΔGBC map ($r_s = 0.72$; p < $10^{-5}$; Spearman rank correlation). This indicates that individual differences are primarily driven by the strength with which subjects exhibit the group-averaged pattern.

To explore individual variation in the model, for each subject, we repeated the two-dimensional grid search over the two gain-modulating parameters. Model-empirical loading for each subject was computed using that subject's ΔGBC map as the target model output. For each subject, we selected the combination of gain modulation parameters that maximized model-empirical loading. We then used those parameters to construct subjects' model ΔGBC maps (*Figure 4—figure supplement 1*). We performed PCA on these subject-specific maps to derive PC1 of ΔGBC variance in the model

(*Figure 4B*). We found that, as in the empirical data, PC1 is topographically aligned with the group-level $\Delta$GBC map ($r_s = 0.87$; p < $10^{-5}$; Spearman rank correlation). Moreover, model PC1 and empirical PC1 exhibited similar spatial topographies ($r_s = 0.63$; p < $10^{-5}$; Spearman rank correlation). These findings reveal a convergence between the dominant modes of neural variation in the data and in the model.

The model PC variance spectrum shows that variation across subjects' simulated $\Delta$GBC maps is almost entirely captured by the two leading model PCs (*Figure 4C*). That model variation lies within a two-dimensional subspace is expected due to our fitting of two free parameters to $\Delta$GBC. We also find topographic alignment between model PC2 and empirical PC2 ($r_s = 0.51$; p < $10^{-5}$; Spearman rank correlation) (*Figure 4D*). However, the empirical PC variance spectrum reveals that the data exhibit approximately five significant modes of variation (*Figure 4C*). The data therefore exhibit three more modes than the model. A linear subspace analysis revealed that 67% of model variance fell within the subspace defined by the data's leading five PCs (see Materials and methods). In other words, we found that the majority of model variation lies within a low-dimensional subspace defined by the principal modes of empirical variation.

With only two significant modes of variation, is the model expressive enough to capture individual differences in LSD-induced alterations of consciousness? As reported by *Preller et al., 2018*, changes in conscious experience were determined psychometrically using the five-dimensional altered states of consciousness (5D-ASC) questionnaire (*Dittrich et al., 2010*), which was administered to study participants 180 min after drug infusion. The short version of the 5D-ASC questionnaire quantifies changes in conscious experience along five distinct experiential dimensions: experiences of disembodiment, elemental imagery, changed meaning of percepts, blissful state, and spiritual experience – each of which is profoundly altered under LSD (*Figure 4E*). We performed a neurobehavioral linear regression analysis to find patterns of altered GBC that were predictive of experiential effects. Specifically, for each (parcel, experiential dimension) pair, we performed a linear regression across subjects, such that each sample in the regression between $\Delta$GBC in parcel $i$ (the predictor variable) and change along experiential dimension $j$ (the target variable) corresponds to a ($\Delta$GBC$_i$, $\Delta$Score$_j$) pair for one subject.

For each experiential dimension, we performed one regression per parcel and aggregated the linear regression coefficients across all parcels to create a spatial brain map (*Figure 4F*). We henceforth refer to these maps as 'experiential regression maps'. By construction, experiential regression maps reflect $\Delta$GBC patterns that predict individuals' change in experience. For instance, if a subject's $\Delta$GBC map strongly resembles the Meaning regression map (*Figure 4F*), then we expect that subject to have reported a relatively strong change in the meaning of percepts. How consistent are these experiential regression maps across experiential dimensions? We found evidence for a single dominant experiential regression map pattern (*Figure 4G*).

Returning to our original question – do modes of model variation have experiential relevance? – we compared PC1 maps (which characterize individual variation) to experiential regression maps (which are defined by their experiential relevance) (*Figure 4G*). We find notable correlations of model PC1 with the Meaning ($r_s = 0.38$; p < $10^{-5}$; Spearman rank correlation) and Disembodiment ($r_s = 0.61$; p < $10^{-5}$; Spearman rank correlation) experiential regression maps. Most strikingly, however, we find that the model PC1 map strongly resembles the dominant experiential regression map pattern ($r_s = 0.65$; p < $10^{-5}$; Spearman rank correlation), which is defined as the first PC of the experiential regression maps. This indicates that the model captures patterns of functional individual variation that are linked to individual differences in pharmacologically induced changes in experience.

## Discussion

In this study we integrated transcriptomic mapping into a biophysically based model of large-scale cortical dynamics to investigate the circuit mechanisms underlying LSD-induced changes in cortical GBC topography. Recently, our group has shown that LSD induces experientially relevant changes in GBC through its agonist activity at the 5-HT$_{2A}$ receptor (*Preller et al., 2018*). Here, we found that the $\Delta$GBC topography was captured in silico when LSD's effect on molecular signaling was modeled as a modulation of neuronal gain, mediated by 5-HT$_{2A}$ receptor agonism, with preferential impact on pyramidal neurons. Moreover, our findings were specifically attributable to the topography of the HTR2A expression map – our transcriptomic proxy measure of the spatial distribution of 5-HT$_{2A}$

receptors. Finally, we showed that the model has enough expressivity to fit individual subjects and exhibits patterns of variation that predict drug-induced alterations of consciousness. This work establishes a framework for linking molecular-level manipulations to salient changes in brain function by integrating transcriptomics, biophysical modeling, and pharmacological neuroimaging.

Our findings complement a recent modeling study by *Deco et al., 2018*, which showed that a gain-modulatory mechanism captures changes in global spatiotemporal properties of group-level neural dynamics under LSD but did not examine cortical topographic effects as studied here. The authors of that study characterized changes in the distribution of pairwise correlations between sliding-window dynamic FC matrices, or FC dynamics (FCD) (*Hansen et al., 2015*). FCD and other global statistical metrics have revealed interesting dynamical consequences of gain modulation in simulated neural systems (*Shine et al., 2018*; *Deco et al., 2018*; *Li et al., 2019*; *Pfeffer et al., 2020*). However, changes in global metrics may be driven by spatially non-specific effects and by non-neuronal components or artifacts, particularly in imaging studies of serotonergic psychedelics (*Lewis et al., 2017*; *Preller et al., 2018*; *Vollenweider and Preller, 2020*). Our findings here demonstrate that gain modulation is a molecular mechanism that explains the specific spatial topography of LSD's functionally disruptive effects. This key result illustrates how our proposed modeling framework could potentially be used in future studies to inform the development of therapeutics that precisely target pathological brain circuits while minimizing off-target effects.

Whereas prior modeling work explored the effects of modulating neural gain uniformly across cell types (*Deco et al., 2018*), our study investigated the cell-type specificity of LSD's effects. Our finding that the best model fits are associated with elevation of *E/I* ratio and gain modulation that is biased toward excitatory populations suggests that LSD preferentially targets pyramidal neurons. This finding converges with multiple lines of experimental evidence in humans (*Pazos et al., 1987*; *Hall et al., 2000*), monkeys (*Jakab and Goldman-Rakic, 1998*), and rats (*Willins et al., 1997*; *Santana et al., 2004*), which show that 5-HT$_{2A}$ receptors are preferentially found on apical dendrites of cortical pyramidal neurons. Moreover, 5-HT$_{2A}$ receptor agonists have been shown to induce substantial increases in cortical pyramidal neuron firing rates (*Martín-Ruiz et al., 2001*; *Puig et al., 2003*; *Lladó-Pelfort et al., 2018*). State-of-the-art single-cell RNA sequencing data across multiple cortical areas in humans also reveals that HTR2A expression is significantly elevated in excitatory cell types, relative to inhibitory cell types (*Figure 1—figure supplement 1*).

Most prior models of psychedelics' pharmacological effects in humans have been fitted to group-level statistical measures (*Deco et al., 2018*; *Preller et al., 2019*; *Kringelbach et al., 2020*; *Herzog et al., 2020*; *Pfeffer et al., 2020*). Here, we have shown that by fitting to individual subjects, the model can capture experientially relevant modes of neural variation. Our model fits the majority of subjects well (*Figure 4—figure supplement 1*). However, the number of subjects included in the empirical data collection supporting this study was not statistically well powered for individual differences predictive analyses, and several subjects were too idiosyncratic in their neural responses for us to accurately predict individuals' experiential responses to LSD. Pharmacological modeling studies informed by other imaging modalities and powered by larger sample sizes are needed to fully characterize the predictive power and limitations of this approach.

LSD primarily mediates its effects on the human brain through 5-HT$_{2A}$ receptors (*Preller et al., 2018*; *Holze et al., 2021*), but LSD also acts as an agonist at other serotonergic and dopaminergic receptor sites (*Nichols, 2004*; *Passie et al., 2008*). In particular, LSD exhibits high affinity for 5-HT$_{1A}$ receptor subtypes, which function as inhibitory autoreceptors in the raphe nuclei and induce postsynaptic membrane hyperpolarization in limbic areas (*Filip and Bader, 2009*). Thus, 5-HT$_{2A}$ and 5-HT$_{1A}$ receptors mediate opposing influences on membrane excitability. Interestingly, this bidirectionality appears consistent with the 'anti-psychedelic' properties of partial 5-HT$_{1A}$ receptor agonists in humans (*Pokorny et al., 2016*). In addition, 5-HT$_{2A}$ receptors are found in all cortical laminae (*Jakab and Goldman-Rakic, 1998*; *Burnet et al., 1995*), often colocalized with 5-HT$_{1A}$ receptors on pyramidal neurons (*Araneda and Andrade, 1991*; *Amargós-Bosch et al., 2004*). 5-HT$_{2A}$ receptors are also found on GABAergic interneurons, though relatively infrequently (*Willins et al., 1997*; *Santana et al., 2004*).

Functional neuroimaging-derived measures including GBC are sensitive to the global signal (GS), which represents shared variation across brain regions and may partially reflect non-neuronal physiological, movement- and scanner-related artifacts (*Murphy et al., 2013*; *Murphy and Fox, 2017*). In particular, GS artifacts exhibit dramatic differences in clinical populations and following

pharmacological manipulations (*Driesen et al., 2013*; *Yang et al., 2014*; *Power et al., 2017*; *Yang et al., 2017*; *Lewis et al., 2017*). The data processing decision to include or forego GSR therefore has impact on findings in studies of psychedelics (*Preller et al., 2018*; *Preller et al., 2020*): it has been shown that results frequently do not replicate across samples when GSR is not performed (*Tagliazucchi et al., 2016*; *Preller et al., 2018*; *Müller et al., 2017*; *Preller et al., 2020*), while studies that use GSR have reported replicable findings (*Preller et al., 2018*; *Preller et al., 2020*). Here, we found that multiple measures of model-empirical similarity were significantly improved when GSR was performed (*Figure 2—figure supplement 1*), providing further support for its use. However, use of GSR remains debated (*Fox et al., 2009*; *Saad et al., 2012*), and in general there is no single 'right' way to process resting-state data (*Murphy and Fox, 2017*). In principle, it may be possible to circumvent some complications introduced by GSR by using a combination of spatial and temporal ICA-based de-noising (*Glasser et al., 2018*), but this has not yet been tested in pharmacological fMRI studies.

In this study we leveraged recent advances in generative null modeling to establish statistical benchmarks while controlling for the confounding influence of spatial autocorrelation in analyses of brain maps (*Burt et al., 2020*). Spatial autocorrelation occurs ubiquitously in empirical brain maps and can dramatically inflate p-values in analyses of both cortical and subcortical brain maps (*Burt et al., 2020*). Importantly, however, maps of brain features often exhibit the most marked differences across (as opposed to within) neuroanatomical structures. For instance, although gene expression profiles exhibit characteristic hierarchical patterns of intra-cortical variation (*Burt et al., 2018*), cortico-cortical gene expression variance is small relative to cortico-subcortical variance (*Hawrylycz et al., 2015*). In addition to spatial autocorrelation within brain structures, these sharp distinctions between brain structures (such as cortex vs. thalamus) introduce additional bias into spatially naive permutation tests performed at the whole-brain level (e.g., as used by *Deco et al., 2018*).

We used gene expression maps, derived from the AHBA, as proxy correlates of receptor densities (*Hawrylycz et al., 2012*; *Hawrylycz et al., 2015*; *Burt et al., 2018*). *Carlyle et al., 2017* found that variation in gene expression levels across human brain regions generally well captured regional variation in protein expression levels. Positron emission tomography (PET) is a neuroimaging modality which can measure regional variation in receptor availability in vivo. *Beliveau et al., 2017* directly compared PET maps for serotonin receptors and found good alignment with gene expression levels from the AHBA. A key limitation of using PET-derived maps for model simulations of pharmacology is that for many receptors of interest, PET mapping is not yet possible due to lack of developed radioligands (*Pike, 2016*). Our study demonstrates proof of concept that transcriptomic mapping can inform large-scale models of pharmacological neuroimaging.

Our parsimonious computational model of LSD's effects in cortex treats excitatory and inhibitory neurons as statistical ensembles, providing a description of neural dynamics at a level of resolution and complexity appropriate for proof-of-concept comparisons with BOLD neuroimaging data. However, future work which expands, constrains, and tests biophysical models of pharmacology will be critical to address a number of open questions that remain. Neuromodulation and disease processes may predominately influence spatiotemporal properties within rather than across areas, suggesting the need for multi-scale models that go beyond descriptions of brain areas in aggregate. Because our model is defined at the level of cortical parcels, it cannot speak to changes in connectivity that occur over smaller spatial scales, particularly among neurons within the parcels themselves. Our findings indicate that this coarse dynamical description is sufficient to capture regional GBC differences, but future work that goes beyond regional mean-field modeling may be needed to fully resolve the fine-grained effects of pharmacology on within- and between-region FC. Cell-type specific pharmacological effects can also be further investigated by multi-compartment models or models which include multiple interneuron subtypes. Cell-type specific effects could be introduced into such models by integrating bulk transcriptomics with single-cell RNA-sequencing to inform the joint distribution of pharmacological targets across distinct areas and cell types, respectively (*Lake et al., 2016*; *Burt et al., 2018*). Moreover, while 5-HT$_{2A}$ receptors are predominately found in cortex (*Figure 1—figure supplement 1A–C*), future model extensions with subcortical structures can incorporate the 5-HT$_{2A}$ receptor-rich claustrum (*Hall et al., 2000*), which may play a role in mediating the effects of serotonergic psychedelic drugs (*Barrett et al., 2020*). Comparing pharmacological models to data from multiple imaging modalities may also generate new and complementary insights: for instance,

the mechanisms underlying LSD-induced changes in broadband oscillatory power (*Muthukumaraswamy et al., 2013*; *Carhart-Harris et al., 2016*) and increased neural signal diversity (*Schartner et al., 2017*) remain unclear.

Recent years have experienced a resurgence of clinical interest in the use of psychedelics as therapeutics for the treatment of mood disorders, alcohol and substance abuse, and end-of-life distress in terminally-ill patients (*Carhart-Harris and Goodwin, 2017*; *Nichols et al., 2017*; *Vollenweider and Preller, 2020*). The common therapeutic mechanism of serotonergic psychedelic drugs remains unclear, although multiple lines of evidence now point to 5-HT$_{2A}$ receptor-mediated glutamate release (*Vollenweider and Kometer, 2010*; *Mason et al., 2020*; *Vollenweider and Preller, 2020*). Spatial gradients in drug targets, such as serotonergic receptor subtypes, can be approximated using the topography of gene transcripts – particularly in the case of receptors for which the field lacks suitable PET radioligands. Studies that leverage transcriptomic mapping (*Grandjean et al., 2021*) and regionally heterogeneous modeling (*Demirtaş et al., 2019*) will further elucidate the mechanisms underlying the complex functional signatures of pharmacologically induced neuromodulatory effects (*Pfeffer et al., 2020*; *Alamia et al., 2020*).

## Materials and methods

### Structural neuroimaging data

Group-averaged left-hemispheric SC matrices were constructed from diffusion MRI data using probabilistic tractography for 339 unrelated subjects from the HCP 900-subject data release (*Van Essen et al., 2013*). SC matrices were parcellated into 180 areas using the HCP's MMP1.0 (*Glasser et al., 2016*). Diagonal elements of the SC matrix were set identically to zero, as the dynamical model (described below) explicitly includes self-coupling terms. Moreover, the SC matrix was row-wise normalized such that the total long-range inputs to each node were normalized. This normalization procedure instantiates the assumption that each local microcircuit receives a balance of local and long-range inputs. The CAB-NP was used to determine parcels' functional network assignments (*Ji et al., 2019*). Pairwise inter-parcel geodesic distance matrices, which were used for constructing spatial autocorrelation-preserving surrogate maps (*Burt et al., 2020*), were computed by averaging over surface-based distances between grayordinate vertices in each pair of parcels $i$ and $j$, where surface-based distances were computed across the left-hemisphere midthickness surface in the HCP atlas.

### Pharmacological fMRI data

A complete description of data collection and pre-processing procedures was reported in *Preller et al., 2018*. Briefly, fMRI data derived from a double-blind, placebo-controlled, and within-subject study design, ensuring that each subject served as their own control. Twenty-five healthy human participants received either (i) placebo or (ii) LSD (100 µg po). Resting-state scans were collected 75 min following drug administration. fMRI data were obtained and processed following HCP-compliant acquisition standards (*Glasser et al., 2013*). Behavioral measures were derived from the short version of the 5D-ASC questionnaire (*Dittrich et al., 2010*), which includes 45 items that comprise five different scales: spiritual experience, blissful state, disembodiment, elementary imagery, and changed meaning of percepts. The 5D-ASC questionnaire was administered 180 min following drug infusion. One subject was excluded from analysis due to failure in registration caused by an improper head position. Thus, 24 study participants were analyzed in this study.

### Transcriptomic data

The AHBA is a publicly available transcriptional atlas of microarray data containing samples from hundreds of neuroanatomical structures in six normal post-mortem human brains. Microarray expression data and all accompanying metadata were downloaded from the AHBA (http://human.brain-map.org) (*Hawrylycz et al., 2012*; *Hawrylycz et al., 2015*). An exhaustive description of our pre-processing procedure was reported in *Burt et al., 2018*. Briefly, cortical microarray data in volumetric space were mapped onto subjects' native two-dimensional cortical surfaces by minimizing three-dimensional Euclidean distance between microarray samples and grayordinate vertex coordinates. For parcels that were not directly sampled, we performed a surface-based interpolation of sample expression values using a Voronoi diagram approach to create a dense gene expression map (at the

level of grayordinates); this interpolated map was then parcellated by averaging across grayordinate vertices in each parcel. One representative microarray probe was selected for each unique gene, and gene expression profiles for selected gene probes were z-scored. Finally, group-averaged expression profiles were computed for genes whose expression profiles were reliable in at least four of the six subjects. These steps yielded group-averaged gene expression values across 180 left-hemispheric cortical areas.

Single-nucleus transcriptomic data that quantify the expression of HTR2A across distinct cell types in human cortex was obtained from the Allen Brain Institute's Human Multiple Cortical Areas SMART-seq database (https://portal.brain-map.org/atlases-and-data/rnaseq/human-multiple-cortical-areas-smart-seq). We downloaded the CSV file containing 'Gene expression by cluster, trimmed means' and aggregated columns that included either 'Exc' or 'Inh'.

## Modeling large-scale neural dynamics

We adapted the biophysically based large-scale computational model described in *Deco et al., 2014*. This model reduces the complexity and the number of local microcircuit parameters in a spiking neural network model using a dynamical mean-field approach (*Wong and Wang, 2006*). By leveraging a statistical description of ensemble neural activity and exploiting the long time constants of NMDA receptors, the model reduces a high-dimensional spiking neural network to a computationally tractable two-dimensional dynamical system. Each node in the model comprises recurrently coupled excitatory (*E*) and inhibitory (*I*) populations, the dynamics of which are described via coupled non-linear stochastic differential equations. Time-varying activity of the excitatory and inhibitory synaptic currents in brain region $i$, denoted respectively by $I_i^E$ and $I_i^I$, is defined as

$$
\begin{aligned}
I_i^E(t) &= W^E I_b + w^{EE} S_i^E(t) + GJ\Sigma_j C_{ij} S_j^E(t) - w^{EI} S_i^I(t) \\
I_i^I(t) &= W^I I_b + w^{IE} S_i^E(t) - S_i^I(t)
\end{aligned}
\tag{1}
$$

where $I_b$ is constant background input current; $S$ is the synaptic gating variable, that is, the fraction of open channels; $G$ is a global conductance parameter which scales the strengths of long-range connections; $J$ is the effective NMDA conductance; $C$ is the SC matrix; and remaining parameters are constants. Synaptic current $I^p$ for population $p \in \{E, I\}$ is transformed into firing rate $r^p$ via the transfer function

$$
r_i^p(t) = \phi(I_i^p(t)) = \frac{a^p I_i^p(t) - b^p}{1 - e^{-d^p(a^p I_i^p(t) - b^p)}},
\tag{2}
$$

where $a$, $b$, and $d$ are cell-type specific parameters governing the form of the input-output relation. In particular, note that parameter $a$, which regulates the slope of the transfer function, corresponds to the neural gain parameter. Note that while this function is unbounded and therefore does not saturate, we confirmed that the node-averaged firing rates in the gain-modulated model (which do not exceed ~15 Hz throughout the parameter sweep in *Figure 2A*) remain in a neurobiologically plausible firing rate regime where this approximation of the F-I curve does not break down.

Finally, the time evolution for synaptic gating variable $S$ is given by

$$
\begin{aligned}
\frac{dS_i^E(t)}{dt} &= -\frac{S_i^E(t)}{\tau^E} + (1 - S_i^E(t))\gamma r_i^E(t) + \sigma\nu_i(t) \\
\frac{dS_i^I(t)}{dt} &= -\frac{S_i^I(t)}{\tau^I} + r_i^I(t) + \sigma\nu_i(t)
\end{aligned}
\tag{3}
$$

where $\tau$ is a synaptic time constant, $\gamma$ is a kinetic parameter, and $\sigma$ is the standard deviation of the stochastic Gaussian input noise $\nu$ (*Wong and Wang, 2006*; *Deco et al., 2014*). Following *Deco et al., 2014*, simulated synaptic covariance matrices were approximated analytically by linearizing *Equations 1* to 3 around the dynamical system's stable fixed point.

Neural gain modulation was implemented in the model by introducing a parametric rescaling of gain parameter $a$ for each cell type, in proportion to regional gene expression levels of HTR2A:

$$
\begin{aligned}
a_i^E &= h_i(1 + \delta^E)a^E \\
a_i^I &= h_i(1 + \delta^I)a^I
\end{aligned}
\tag{4}
$$

where $h_i$ denotes the expression level of HTR2A in brain region $i$, $\delta^E$ and $\delta^I$ regulate the strength of the perturbation applied to excitatory and inhibitory cell populations, respectively. Thus, two-dimensional parameter sweeps reported in this study, for instance in *Figure 2A*, correspond to sweeps over free parameters $\delta^E$ and $\delta^I$. Where brain maps (gene expression maps as well as surrogate maps) were used to modulate gain, they were first linearized using the error function (following *Demirtaş et al., 2019*).

Following *Deco et al., 2014*, we implemented feedback inhibition control (FIC) in the model such that the firing rates $r_i^E$ were constrained to be 3 Hz prior to neural gain modulation. This is accomplished by tuning the weight $w^{EI}$, which regulates the strength of local inhibitory-to-excitatory feedback, within each simulated brain region. Crucially, we note that FIC was calculated prior to but not following gain modulation, under the assumption that FIC represents neurobiological self-regulatory processes which balance neural activity in the brain on timescales much longer than the characteristic timescales of transient pharmacological manipulations.

## Modeling the hemodynamic response

To facilitate model comparisons with empirical fMRI data, excitatory synaptic activity within each brain region (given by $S^E$ in *Equation 3*) was transformed to a BOLD signal using the Balloon-Windkessel model (*Friston et al., 2003*). In this model, the hemodynamic response obeys the following coupled system of equations:

$$\frac{dx_i(t)}{dt} = S_i^E(t) - kx_i - \gamma(f_i(t) - 1) \tag{5}$$

$$\frac{df_i(t)}{dt} = x_i(t) \tag{6}$$

$$\tau \frac{dv_i(t)}{dt} = f_i(t) - v_i^{\frac{1}{\alpha}}(t) \tag{7}$$

$$\tau \frac{dq_i(t)}{dt} = \frac{f_i(t)}{\rho} \left[ 1 - (1-\rho)^{\frac{1}{f_i(t)}} \right] - q_i \left( v_i^{\frac{1-\alpha}{\alpha}}(t) \right) \tag{8}$$

where $x$ denotes the vasodilatory signal, $f$ the blood inflow, $v$ the blood volume, and $q$ the deoxyhemoglobin content; and parameters $\rho$, $\tau$, $\kappa$, $\gamma$, and $\alpha$ are the resting oxygen extraction fraction, hemodynamic transit time, rate of signal decay, rate of flow-dependent elimination, and the Grubb's exponent, respectively (*Friston et al., 2003*). In turn, the corresponding BOLD signal $y$ is calculated as

$$y_i(t) = V_0 \left[ k_1(1-q_i(t)) + k_2 \left(1 - \frac{q_i(t)}{v_i(t)}\right) + k_3(1-v_i(t)) \right] \tag{9}$$

where $V_0$ is the resting blood volume fraction. The three dimensionless magnetic field strength-dependent parameter values $k_1$, $k_2$, and $k_3$ were derived for a magnetic field strength of 3 T using Appendix A of *Heinzle et al., 2016*; all other hemodynamic parameter values were taken from *Obata et al., 2004*. Simulated BOLD covariance matrices were derived by linearizing these equations and then algebraically transforming the linearized covariance matrix, using a procedure which we previously reported in *Demirtaş et al., 2019*. Specifically, we semi-analytically derive the covariance matrix $P$ of the dynamical system by numerically solving the Lyapunov equation:

$$AP + PA^T + Q_N = 0 \tag{10}$$

where $A$ is the Jacobian matrix and $Q_N$ is the noise covariance matrix. Note that this expression is solved for the full six-dimensional dynamical system which includes two synaptic variables, $S^E$ and $S^I$, as well as four hemodynamic state variables $x$, $f$, $v$, and $q$. The linearized BOLD covariance matrix is then given by

$$P^{\text{BOLD}} = KPK^{\dagger} \tag{11}$$

where $K$ is a matrix of partial derivatives of the BOLD signal with respect to the six dynamical variables (**Demirtaş et al., 2019**).

Values for all static parameters in the model are provided in *Table 1*.

## Modeling fitting

Tuning the model parameters consisted of two key sequential steps. First, the dynamical operating point of the model was calibrated such that (prior to gain modulation) the model optimally captured empirical FC derived from the placebo-condition fMRI scan. To this end, we performed a one-dimensional grid search over the global coupling parameter $G$, which linearly scales all long-range interactions between nodes (*Equation 1*). The grid search was over the range [0.01, 0.85] with a step size of 0.01. For each value, we solved for the stable fixed point of the system and then computed the linearized BOLD covariance matrix around that point. From the BOLD covariance matrix, we computed the FC matrix using the transformation from covariance to Pearson correlation coefficient. We computed the Spearman rank correlation between the $N = 180(180 - 1)/2$ upper-triangular elements of the model FC matrix and the placebo-condition empirical FC matrix. At the group level (i.e., using the group-averaged empirical FC), this step yielded an optimal global coupling value of $G = 0.85$, which resulted in a Spearman rank correlation of $r_s = 0.45$. This value of $G$ was used for all group-averaged analyses in this study. We note that if the upper end of the range was extended beyond $G = 0.85$, the maximal fit to the group-level data occurred at $G = 0.89$. However, at this value the fit increases only marginally ($\delta r_s < 10^{-3}$), while the system is then positioned so close to a critical bifurcation that even modest modulations of gain will dynamically destabilize the system. This class of computational models is known to produce optimal fits near the edge of stability (see, e.g., **Deco et al., 2013**).

**Table 1.** Fixed parameter values used in synaptic and hemodynamic equations.

| | Excitatory populations | Inhibitory populations |
|---|---|---|
| Synaptic model parameters | | |
| $I_b$ | 0.382 nA | – |
| $J$ | 0.15 nA | – |
| $\gamma$ | 0.641 | – |
| $W^E$ | 1.0 | – |
| $\tau^E$ | 0.1 s | – |
| $a^E$ | 310 nC$^{-1}$ | – |
| $b^E$ | 125 Hz | – |
| $d^E$ | 0.16 s | – |
| $W^I$ | – | 0.7 |
| $\tau^I$ | – | 0.01 s |
| $a^I$ | – | 615 nC$^{-1}$ |
| $b^I$ | – | 177 Hz |
| $d^I$ | – | 0.087 s |
| Hemodynamic model parameters | | |
| $\rho$ | 0.34 | – |
| $\alpha$ | 0.32 | – |
| $V_0$ | 0.02 | – |
| $\gamma$ | 0.41 s$^{-1}$ | – |
| $\kappa$ | 0.65 s$^{-1}$ | – |
| $k_1$ | 3.72 | – |
| $k_2$ | 0.53 | – |
| $k_3$ | 0.53 | – |

After calibrating the system, next we performed a two-dimensional grid search over model parameters $\delta E$ and $\delta I$, which determine the strength of excitatory and inhibitory gain modulation, respectively. For each combination of parameters, we updated the neural gain equations (*Equation 5*) in the model. We solved for the new fixed point and computed the linearized BOLD covariance matrix around that point. FC and GBC were then computed as described above. The difference between model GBC maps (expressed as Fisher Z-values) – perturbed minus baseline – yielded a model ΔGBC map. We then computed the model-empirical loading between the model ΔGBC map and the empirical ΔGBC map (LSD minus placebo). As illustrated in the heatmaps, the grid search for the gain modulation parameters was over the range [0, 0.03] with a step size of 0.003. Gain parameters that maximized model-empirical loading were then used for all analyses involving a model ΔGBC map, either at the group or subject level.

## Global brain connectivity

Empirical GBC was computed using in-house Matlab tools for all grayordinates in the brain using an HCP-harmonized version of the FreeSurfer software (*Fischl et al., 2002*; *Cole et al., 2011*; *Anticevic et al., 2013*; *Anticevic et al., 2014*), as reported in *Preller et al., 2018*. Briefly, for each subject, we computed the fMRI time-series correlation between pairs of grayordinates, thereby constructing an FC matrix. Off-diagonal FC matrix elements were Fisher Z-transformed and then averaged across rows to compute GBC values. Maps were then parcellated using the HCP's MMP1.0 (*Glasser et al., 2016*). This yielded a parcellated GBC map for each subject where each parcel's value represents the mean FC of that region to all other regions. Parcellated GBC maps were averaged across subjects to construct the group-averaged GBC map. Similarly, to compute model GBC maps, we used the model to generate a parcellated BOLD FC matrix. Off-diagonal FC matrix elements were Fisher Z-transformed and then averaged across rows to compute GBC values. All illustrated GBC map values (empirical and simulated) are Fisher Z-values.

## Global signal regression

GSR was performed on empirical fMRI-derived time-series data using widely adopted approaches (*Anticevic et al., 2013*; *Cole et al., 2011*), as described in *Preller et al., 2018*. GSR was also performed in the model, as GSR not only removes artifactual signal components but neuronal signal as well. In the model, however, we performed GSR directly on the analytically approximated BOLD covariance matrices using the following approach. Let $y_i$ be the BOLD signal in brain region $i$, and let $x$ be the GS – that is, the mean gray-matter BOLD signal across regions. Note that $x$ and $y$ are both implicit functions of time. We construct a linear regression model of the form

$$\hat{y}_i = \alpha_i + \beta_i x, \tag{12}$$

where the residual signal following GSR is given by

$$\epsilon_i \equiv y_i - \hat{y}_i. \tag{13}$$

What we seek is an expression for the covariance between GS-regressed BOLD signal residuals in regions $i$ and $j$, that is,

$$\mathrm{Cov}(\epsilon_i, \epsilon_j) = \left\langle (\epsilon_i - \overline{\epsilon_i})(\epsilon_j - \overline{\epsilon_j}) \right\rangle, \tag{14}$$

where $\overline{\epsilon}$ denotes the temporal average of $\epsilon$. Substituting *Equations 12 and 13*, this expression can be rewritten as

$$\mathrm{Cov}(\epsilon_i, \epsilon_j) = \begin{aligned} &\left\langle (y_i - \overline{y_i}), (y_j - \overline{y_j}) \right\rangle - \left\langle (y_i - \overline{y_i}), \beta_j(x_j - \overline{x_j}) \right\rangle - \\ &\left\langle \beta_i(x_i - \overline{x_i})(y_j - \overline{y_j}) \right\rangle + \left\langle \beta_i(x_i - \overline{x_i})\beta_j(x_j - \overline{x_j}) \right\rangle. \end{aligned} \tag{15}$$

By manipulating *Equation 12*, $\beta$ can be expressed in terms of covariances:

$$\beta = \frac{\mathrm{Cov}(x, \hat{y})}{\mathrm{Cov}(x, x)} = \frac{\left\langle (x_i - \overline{x_i})(\hat{y}_j - \overline{\hat{y}_j}) \right\rangle}{\left\langle (x_i - \overline{x_i})(x_j - \overline{x_j}) \right\rangle}. \tag{16}$$

Substituting this expression into *Equation 15* and simplifying, we obtain

$$\text{Cov}(\epsilon_i, \epsilon_j) = \langle (y_i - \overline{y_i})(y_j - \overline{y_j}) \rangle - \frac{\langle (y_i - \overline{y_i})(x - \overline{x}) \rangle \langle (y_j - \overline{y_j})(x - \overline{x}) \rangle}{\langle (x - \overline{x})^2 \rangle}. \tag{17}$$

From the definition of GS $x = \sum_i y_i$, *Equation 17* can be rewritten as

$$\text{Cov}(\epsilon_i, \epsilon_j) = \langle (y_i - \overline{y_i})(y_j - \overline{y_j}) \rangle - \frac{\langle (y_i - \overline{y_i})(\sum_k y_k - \sum_k \overline{y_k}) \rangle \langle (y_j - \overline{y_j})(\sum_l y_l - \sum_l \overline{y_l}) \rangle}{\langle (\sum_k y_k - \sum_k \overline{y_k})^2 \rangle}. \tag{18}$$

Collecting the summations and noting that

$$\text{Cov}(y_i, y_j) = \langle (y_i - \overline{y_i})(y_j - \overline{y_j}) \rangle, \tag{19}$$

we obtain

$$\text{Cov}(\epsilon_i, \epsilon_j) = \text{Cov}(y_i, y_j) - \frac{\sum_k \text{Cov}(y_i, y_k) \sum_l \text{Cov}(y_j, y_l)}{\sum_{m,n} \text{Cov}(y_m, y_n)}. \tag{20}$$

*Equation 20* analytically relates the covariance matrix of the raw BOLD signal (right-hand side) to the covariance matrix of GS-regressed residuals (left-hand side). We used this expression to perform GSR on model BOLD covariance matrices. FC matrices (i.e., Pearson correlation matrices) were then computed from GS-regressed BOLD covariance matrices.

## Generative modeling of surrogate brain maps

Spatial autocorrelation-preserving surrogate maps were constructed following *Burt et al., 2020* using the Python-based BrainSMASH toolbox (https://brainsmash.readthedocs.io/). This approach operationalizes spatial autocorrelation in a brain map by computing a variogram, which quantifies variance in the brain map as a function of pairwise distance between areas. To construct each surrogate map, the empirical brain map is iteratively shuffled (to randomize topography), smoothed (to reintroduce spatial autocorrelation), and rescaled (to recover empirical spatial autocorrelation). After performing this procedure, we resampled our surrogate map values from the empirical brain map such that the distribution of values does not change: each surrogate map can then be conceptualized as one random realization of an (approximately) spatial autocorrelation-preserving permutation of the original brain map.

For each surrogate map $\vec{h}$, the surrogate map values $\{h_i\}$ were substituted into *Equation 5* such that the modulation of gain in each region was scaled in proportion to the values of the surrogate map, rather than the empirical HTR2A map. This 'surrogate model' was then used to re-compute the statistic under consideration. Repeating for and aggregating across all surrogate maps yields a null distribution for the expected value of the statistic under the null hypothesis that the outcome of the statistical test is an artifact of spatial autocorrelation, and therefore not unique to a specific spatial topography. Specifically, where we report p-values that derive from a spatial autocorrelation-preserving surrogate map test, we are reporting the proportion of samples in the null distribution which were more extreme than the test statistic.

## Experiential regression maps

To identify characteristic patterns of changes in GBC that predict subject-level changes in 5D-ASC scores, we constructed spatial brain maps of linear regression coefficients (i.e., beta coefficient maps or simply 'beta maps'). To calculate the beta map value in the $i$ th cortical parcel for experiential dimension $d$, we solved for $\beta_i^d$ in the linear regression defined by

$$\overrightarrow{\delta s^d} = \alpha_i^d + \beta_i^d \cdot \overrightarrow{\Delta \text{GBC}_i} \tag{21}$$

where $\delta s^d$ is the change in score $\delta s$ along experiential dimension $d$; $\alpha$ and $\beta$ are constant and linear regression coefficients, respectively; and $\Delta \text{GBC}_i$ is the change in GBC for parcel $i$; and vectors include 24 elements (one per subject).

## Linear decomposition

To define low-dimensional linear subspaces within the 180-dimensional neural state space, we used PCA to determine the dominant spatial modes of variation across different collections of brain maps. We let $\mathbf{X}$ be the $M \times N$ matrix comprised of $M$ brain maps, each represented as an $N$-dimensional vector, where columns of $\mathbf{X}$ have been mean-subtracted. The spatial covariance matrix is then given by the $N \times N$ symmetric matrix $\mathbf{C}$, where

$$\mathbf{C} = \frac{1}{M-1}\mathbf{X}^T\mathbf{X}. \tag{22}$$

The symmetric matrix $\mathbf{C}$ can in general be decomposed according to

$$\mathbf{C} = \mathbf{P}\Lambda\mathbf{P}^T, \tag{23}$$

where $\mathbf{P}$ is an $N \times N$ orthogonal matrix whose $i$ th column $\vec{p}_i$ is an eigenvector of $\mathbf{C}$, and $\Lambda$ is an $N \times N$ diagonal matrix whose $i$ th diagonal element $\lambda_i$ is the eigenvalue associated with eigenvector $p_i$. Eigenvector $\vec{p}_i$ is a PC of $\mathbf{X}$, and the normalized eigenvalue $\lambda_i/\mathrm{Tr}(\Lambda)$ is the fraction of total variance in $\mathbf{X}$ that occurs along $\vec{p}_i$. We will assume without loss of generality that eigenvectors are ordered such that $\lambda_{i+1} \leq \lambda_i$. In other words, we assume that eigenvectors are arranged such that eigenvalues are placed in descending order. Finally, note that if $M<N$, then $N-M+1$ eigenvalues are identically zero (i.e., $\mathbf{C}$ will not be full rank).

To determine which PCs were statistically significant, we performed simple permutation testing. Each row in $\mathbf{X}$ (i.e., each brain map) was randomly permuted to obtain a new matrix $\mathbf{X}'$. The procedure described above was then performed on $\mathbf{X}'$ and the fraction of variance captured per PC (i.e., $\vec{\lambda'}/\mathrm{Tr}(\Lambda')$) was recorded. To construct the gray dashed lines in *Figure 4*, we performed this procedure $N = 1000$ times and then computed the fifth percentile of the resulting null distribution (for each individual PC).

To investigate shared dimensions of variability in the model and in the data, we first performed PCA on both the model and empirical subject-specific $\Delta$GBC maps. Permutation testing revealed that the leading two model PCs and the leading five empirical PCs were statistically significant. We then derived the covariance matrix $\mathbf{C}$ for the $24 \times 180$ data matrix $\mathbf{X}$, comprised of subjects' model $\Delta$GBC maps, per *Equation 22*. Total model variance $V$ was determined by computing the trace of $\mathbf{C}$. Let $\mathbf{P}$ denote the $180 \times 5$ matrix whose columns correspond to the leading five empirical PCs; then we performed a linear transformation of $\mathbf{C}$ into the five-dimensional subspace defined by the columns of $\mathbf{P}$ according to

$$\mathbf{C}' = \mathbf{P}^T\mathbf{C}\mathbf{P}. \tag{24}$$

Finally, we determined the total variance $V'$ contained in the subspace by computing the trace of $\mathbf{C}'$. Then the fraction of model variance that falls within the data's five-dimensional subspace (67%) is given by $V'/V$.

## Data and code availability

A Python-based implementation of the model is provided as an open-access software package, BRAINTRIPS (BRainwide Activity Induced by Neuromodulation via TRanscriptomics-Informed Pharmacological Simulation): https://github.com/murraylab/braintrips (*Burt, 2021*; copy archived at swh:1:rev:2f4c9bb63ec01c40e4374e277d03c7ff28e92713). All processed neuroimaging data needed to reproduce main findings in this study are made publicly available on BALSA (https://balsa.wustl.edu/study/show/5XMxv).

## Acknowledgements

This research was supported by grants from the NIH (R01MH112746, JDM; DP5OD012109, AA; R01MH108590, AA); the Swiss National Science Foundation (P2ZHP1_161626, KHP); the Swiss Neuromatrix Foundation (2015–0103, FXV); the Usona Institute (2015–2056, FXV); the NIAAA (P50AA012870-16, AA and JHK); the NARSAD Independent Investigator Grant (AA), a SFARI Pilot Award (JDM, AA); and the Yale CTSA grant (UL1TR000142 Pilot Award, AA). The funding sources

had no involvement in the study design; nor in the collection, analysis, and interpretation of data; nor in the writing of the manuscript; nor in the decision to submit the manuscript for publication.

## Additional information

### Competing interests

Joshua B Burt: JBB is currently an employee of RBNC Therapeutics. Katrin H Preller: KHP is currently an employee of Hoffmann-La Roche. Jie Lisa Ji: JJ has a consulting agreement with BlackThorn Therapeutics. John H Krystal: JHK has consulting agreements (less than US$10,000 per year) with the following: AstraZeneca Pharmaceuticals, Biogen, Idec, MA, Biomedisyn Corporation, Bionomics, Limited (Australia), Boehringer Ingelheim International, COMPASS Pathways, Limited, United Kingdom, Concert Pharmaceuticals, Inc, Epiodyne, Inc, EpiVario, Inc, Heptares Therapeutics, Limited (UK), Janssen Research \& Development, Otsuka America, Pharmaceutical, Inc, Perception Neuroscience Holdings, Inc, Spring Care, Inc, Sunovion Pharmaceuticals, Inc, Takeda Industries and Taisho Pharmaceutical Co., Ltd. JHK serves on the scientific advisory boards of Bioasis Technologies, Inc, Biohaven Pharmaceuticals, BioXcel Therapeutics, Inc (Clinical Advisory Board), BlackThorn Therapeutics, Inc, Cadent Therapeutics (Clinical Advisory Board), Cerevel Therapeutics, LLC., EpiVario, Inc, Lohocla Research Corporation, PsychoGenics, Inc; is on the board of directors of Inheris Biopharma, Inc; has stock options with Biohaven Pharmaceuticals Medical Sciences, BlackThorn Therapeutics, Inc, EpiVario, Inc and Terran Life Sciences; and is editor of Biological Psychiatry with income greater than $10,000. Alan Anticevic: AA has a consulting agreement with BlackThorn Therapeutics. AA is co-inventor of United States patent 10950327 "Methods and systems for computer-generated predictive application of neuroimaging and gene expression mapping data". John D Murray: JDM has a consulting agreement with BlackThorn Therapeutics. JDM is co-inventor of United States patent 10950327 "Methods and systems for computer-generated predictive application of neuroimaging and gene expression mapping data". The other authors declare that no competing interests exist.

### Funding

| Funder | Grant reference number | Author |
|---|---|---|
| National Institute of Mental Health | R01MH112746 | John D Murray |
| National Institute of Mental Health | DP5OD012109 | Alan Anticevic |
| National Institute of Mental Health | R01MH108590 | Alan Anticevic |
| Swiss National Science Foundation | P2ZHP1\161626 | Katrin H Preller |
| Swiss Neuromatrix Foundation | 2015-0103 | Franz X Vollenweider |
| Usona Institute | 2015-2056 | Franz X Vollenweider |
| National Institute on Alcohol Abuse and Alcoholism | P50AA012870-16 | John H Krystal Alan Anticevic |
| Brain and Behavior Research Foundation | Independent Investigator Grant | Alan Anticevic |
| Simons Foundation | Pilot Award | Alan Anticevic John D Murray |
| National Center for Advancing Translational Sciences | UL1TR000142 | Alan Anticevic |

The funders had no role in study design, data collection and interpretation, or the decision to submit the work for publication.

## Author contributions
Joshua B Burt, Conceptualization, Software, Formal analysis, Investigation, Visualization, Methodology, Writing - original draft, Writing - review and editing; Katrin H Preller, Conceptualization, Data curation, Writing - review and editing; Murat Demirtas, Software, Methodology, Writing - review and editing; Jie Lisa Ji, Data curation, Software; John H Krystal, Conceptualization, Writing - review and editing; Franz X Vollenweider, Resources, Writing - review and editing; Alan Anticevic, Conceptualization, Resources, Software, Funding acquisition, Writing - review and editing; John D Murray, Conceptualization, Supervision, Funding acquisition, Writing - original draft, Project administration, Writing - review and editing

## Author ORCIDs
Joshua B Burt ![ORCID] https://orcid.org/0000-0002-5605-2091
Katrin H Preller ![ORCID] https://orcid.org/0000-0003-0413-7672
Jie Lisa Ji ![ORCID] https://orcid.org/0000-0002-6280-9070
Franz X Vollenweider ![ORCID] https://orcid.org/0000-0001-9053-6164
Alan Anticevic ![ORCID] https://orcid.org/0000-0002-4324-0536
John D Murray ![ORCID] https://orcid.org/0000-0003-4115-8181

## Ethics
Clinical trial registration ClinicalTrials.gov: NCT02451072.
Human subjects: This study did not involve any new data collection. Secondary data analysis was performed on the data set described in our prior publication (Preller et al., 2018, eLife). The Swiss Federal Office of Public Health, Bern, Switzerland, authorized the use of LSD in humans, and the study was approved by the Cantonal Ethics Committee of Zurich (KEK-ZH_No: 2014_0496). The study was registered at ClinicalTrials.gov (NCT02451072).

## Decision letter and Author response
Decision letter https://doi.org/10.7554/eLife.69320.sa1
Author response https://doi.org/10.7554/eLife.69320.sa2

# Additional files

## Supplementary files
• Transparent reporting form

## Data availability
A Python-based implementation of the model is provided as an open-access software package, BRAINTRIPS (BRainwide Activity Induced by Neuromodulation via TRanscriptomics-Informed Pharmacological Simulation): https://github.com/murraylab/braintrips (copy archived at https://archive.softwareheritage.org/swh:1:rev:2f4c9bb63ec01c40e4374e277d03c7ff28e92713). All processed neuroimaging data needed to reproduce main findings in this study are made publicly available on BALSA: https://balsa.wustl.edu/study/show/5XMxv.

The following dataset was generated:

| Author(s) | Year | Dataset title | Dataset URL | Database and Identifier |
|---|---|---|---|---|
| Burt JB, Preller KH, Demirtas M, Ji JL, Krystal JH, Vollenweider FX, Anticevic A, Murray JD | 2021 | Transcriptomics-informed modeling of pharmacological neuroimaging effects of LSD | https://balsa.wustl.edu/study/show/5XMxv | BALSA, 5XMxv |

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
