## [Decision Letter]

**Acceptance summary:**

This paper will be of interest to scientists working on computational modelling of neuroimaging data, and on the neural effects of psychedelic drugs and other pharmacological interventions. The study is well-motivated. The statistical and data analytic methodologies are rigorous and advanced. The with conclusions are well-supported by the presented data. The modelling methodology includes technical innovations that are potentially of broad utility and importance.

**Decision letter after peer review:**

Thank you for submitting your article "Transcriptomics-informed large-scale cortical model captures topography of pharmacological neuroimaging effects of LSD" for consideration by *eLife*. Your article has been reviewed by 3 peer reviewers, and the evaluation has been overseen by a Reviewing Editor and Timothy Behrens as the Senior Editor. The following individuals involved in review of your submission have agreed to reveal their identity: James M Shine (Reviewer #1); John Griffiths (Reviewer #2).

Essential revisions:

None. We like the manuscript and would be happy to publish it tomorrow if you choose. However, we have made some suggestions below that we think would improve the manuscript, which you can implement at your discretion.

*Reviewer #1 (Recommendations for the authors):*

P4 – perhaps better to state that GBC 'can be interpreted' as a measure of functional integration.

P4 – I realise it is passe to ask for citations of one's own work, however the authors may wish to cite a recent review on the use of neural mass models linking structure and function (Shine et al., 2021 in Nature Neuroscience), as the topic of the review is strongly-aligned with the authors approach.

P6 – is it realistic to model gain as a continually increasing function? There is a natural ceiling to how high the firing rate of a neuron can be, which suggests that a sigmoid transfer function might be a more sensible function. The authors could mitigate this concern by confirming that the firing rate of their neural populations is bounded within reasonable limits by other features of their model (e.g., EI balance).

P7 – are the off-diagonal elements of the FC matrix normalized separately for 5HT2A vs. placebo conditions? If so, did the authors first check to determine whether there were systematic differences between 5HT2A and placebo that may have been diminished through normalisation?

P8 – the authors conclude that the model fits reflect the fact that "neural gain is preferentially modulated on excitatory pyramidal neurons", however I wonder whether a more parsimonious description would be that "neural gain is preferentially modulated on excitatory pyramidal neurons, which nonetheless remain in a specific ratio with inhibitory interneurons". Note that this result is further expanded in Figure 2B, but I worry that the interim conclusion may mislead from the final result.

P8 – I really liked the utilisation of other 5HT and DA receptor maps and permutation testing.

*Reviewer #2 (Recommendations for the authors):*

It would be informative to include some discussion of the connectivity normalization choice. The connectivity matrix diagonals are set to zero and each row is re-scaled to unity, which is equivalent to using the Laplacian. The result of this is that every brain region receives an identical total level of input from other regions in the network. This is particularly interesting given the principal metric of interest is (changes in) global brain connectivity (row/column averages of the FC matrix). The equivalent maps for the anatomical connectivity will be uniform, for the reasons detailed above. Could the authors please discuss: what, if any, is the neurobiological, and/or mathematical rationale for this normalization choice, and what are their thoughts on the above considerations.

*Reviewer #3 (Recommendations for the authors):*

I really enjoyed reading this manuscript, it is very well-written and easy to follow. The study appears to be methodologically sound, however, given my lack of direct expertise in the modeling of dynamical systems, I am will refrain from giving more specific comments and/or suggestions for improving the scientific quality of these analyses.

---

## [Author Response]

Reviewer #1 (Recommendations for the authors):P4 – perhaps better to state that GBC 'can be interpreted' as a measure of functional integration.

We now state: “…global brain connectivity (GBC), which is a graph-theoretic statistic that can be interpreted as a measure of functional integration.”

P4 – I realise it is passe to ask for citations of one's own work, however the authors may wish to cite a recent review on the use of neural mass models linking structure and function (Shine et al., 2021 in Nature Neuroscience), as the topic of the review is strongly-aligned with the authors approach.

This new review is now cited.

P6 – is it realistic to model gain as a continually increasing function? There is a natural ceiling to how high the firing rate of a neuron can be, which suggests that a sigmoid transfer function might be a more sensible function. The authors could mitigate this concern by confirming that the firing rate of their neural populations is bounded within reasonable limits by other features of their model (e.g., EI balance).

We now state in the Methods: “Note that while this function is unbounded and therefore does not saturate, we confirmed that the node-averaged firing rates in the gain-modulated model (which do not exceed ∼15Hz throughout the parameter sweep in Figure 2A) remain in a neurobiologically plausible firing-rate regime where this approximation of the F-I curve does not break down.”

P7 – are the off-diagonal elements of the FC matrix normalized separately for 5HT2A vs. placebo conditions? If so, did the authors first check to determine whether there were systematic differences between 5HT2A and placebo that may have been diminished through normalisation?

We now include this clarification: “The location of this regime suggests that neural gain is preferentially modulated on excitatory pyramidal neurons, which nonetheless remain in a specific ratio with inhibitory interneurons.”

P8 – the authors conclude that the model fits reflect the fact that "neural gain is preferentially modulated on excitatory pyramidal neurons", however I wonder whether a more parsimonious description would be that "neural gain is preferentially modulated on excitatory pyramidal neurons, which nonetheless remain in a specific ratio with inhibitory interneurons". Note that this result is further expanded in Figure 2B, but I worry that the interim conclusion may mislead from the final result.P8 – I really liked the utilisation of other 5HT and DA receptor maps and permutation testing.Reviewer #2 (Recommendations for the authors):It would be informative to include some discussion of the connectivity normalization choice. The connectivity matrix diagonals are set to zero and each row is re-scaled to unity, which is equivalent to using the Laplacian. The result of this is that every brain region receives an identical total level of input from other regions in the network. This is particularly interesting given the principal metric of interest is (changes in) global brain connectivity (row/column averages of the FC matrix). The equivalent maps for the anatomical connectivity will be uniform, for the reasons detailed above. Could the authors please discuss: what, if any, is the neurobiological, and/or mathematical rationale for this normalization choice, and what are their thoughts on the above considerations.

We now state in the Methods: “Diagonal elements of the SC matrix were set identically to zero, as the dynamical model (described below) explicitly includes self-coupling terms. Moreover, the SC matrix was row-wise normalized such that the total long-range inputs to each node were normalized. This normalization procedure instantiates the assumption that each local microcircuit receives a balance of local and long-range inputs.”